# SAR131675, a VEGRF3 Inhibitor, Modulates the Immune Response and Reduces the Growth of Colorectal Cancer Liver Metastasis

**DOI:** 10.3390/cancers14112715

**Published:** 2022-05-31

**Authors:** Katrina A. Walsh, Georgios Kastrappis, Theodora Fifis, Rita Paolini, Christopher Christophi, Marcos V. Perini

**Affiliations:** 1Department of Surgery, The University of Melbourne, Austin Health, Lance Townsend Building, Level 8, 145 Studley Road, Heidelberg, VIC 3084, Australia; kawalsh@unimelb.edu.au (K.A.W.); g.kastrappis@student.unimelb.edu.au (G.K.); tfifis@unimelb.edu.au (T.F.); c.christophi@unimelb.edu.au (C.C.); 2Melbourne Dental School, The University of Melbourne, Grattan Street, Parkville, VIC 3010, Australia; rita.paolini@unimelb.edu.au

**Keywords:** colorectal cancer, liver metastases, tyrosine kinase, VEGFR3, lymphangiogenesis, myeloid derived suppressor cells, immune system

## Abstract

**Simple Summary:**

Colorectal cancer most often metastasizes to the liver, and in most cases, it is unresectable at diagnosis. New treatment options targeting specific cancer characteristics are needed and are currently being explored. Herein we looked at the use of a selective VEGFR-3 tyrosine kinase inhibitor, SAR131675, as an anti-tumor agent in a mouse model of colorectal liver metastasis. We found that SAR131675 dramatically reduced tumor growth and changed the immune response within the tumor and the surrounding liver, suggesting the use of SAR131675 as an adjuvant therapy for colorectal liver metastasis.

**Abstract:**

Most patients with colorectal cancer (CRC) develop metastases, predominantly in the liver (CLM). Targeted therapies are being investigated to improve current CLM treatments. This study tested the effectiveness of SAR131675, a selective VEGFR-3 tyrosine kinase inhibitor, to inhibit CLM in a murine model. Following intrasplenic induction of CLM, mice were treated daily with SAR131675. Tumor growth and immune infiltrates into tumor and liver tissues were assessed at 10-, 16- and 22-days post tumor induction by stereology, IHC and flow cytometry. SAR151675 treatment significantly reduced tumor burden and F4/80^+^ macrophages in the liver tissues. Analysis of immune cell infiltrates in liver showed tissue that at day 22, had the proportion of CD45^+^ leukocytes significantly reduced, particularly myeloid cells. Analysis of myeloid cells (CD11b^+^ CD45^+^) indicated that the proportion of F4/80^−^ Ly6C^low^ was significantly reduced, including a predominate PD-L1^+^ subset, while CD3^+^ T cells increased, particularly CD8^+^ PD1^+^, reflected by an increase in the CD8^+^:CD4^+^ T cell ratio. In the tumor tissue SAR11675 treatment reduced the predominant population of F4/80^+^ Ly6C^lo^ and increased CD4^+^ T cells. These results suggest that SAR131675 alters the immune composition within tumor and the surrounding liver in the later stages of development, resulting in a less immunosuppressive environment. This immunomodulation effect may contribute to the suppression of tumor growth.

## 1. Introduction

The majority of colorectal cancer deaths are due to tumor metastasis which occur most often in the liver. Surgical resection, the only potential curative treatment, is possible in approximately 40% of cases, leaving most patients the option of palliative chemotherapy. Current standard chemotherapy treatments for unresectable colorectal cancer liver metastasis (CLM) have increased patient survival time from 9 to 20 months. However, these treatments are not specific, have severe side effects and are considered only palliative [1]. Research in new treatments has focused on tumor specific attributes that are critical for tumor progression including upregulated growth factors, mutations that allow constitutive activation of pathways involved in cell growth, and more recently in rebalancing the immune response towards tumor inhibition. Vascular endothelial growth factors (VEGF) and their tyrosine kinase (TK) receptors (VEGFRs) are expressed in primary and metastatic colorectal adenocarcinoma and are also found expressed in the lymph nodes of patients with recurrent disease [2,3]. They are important modulators of neovascularization and have critical roles in tumor vascularization. Signaling through VEGF-A/VEGFR1 and 2 stimulates tumor growth and dissemination. VEGF-A/VEGFR2 signaling is a central activator of tumor angiogenesis and endothelial permeability, while VEGF-C/VEGFR3 signaling is important in lymphangiogenesis [4]. Neovascularization in adults is limited to tissues with high turnover, therefore targeting angiogenic/lymphangiogenic processes for cancer therapy was hoped to be largely tumor specific [5]. Extensive animal studies and clinical trials using monoclonal antibodies or small molecular weight inhibitors, often targeting multiple angiogenic/lymphangiogenic TKs, resulted in a number of these treatments gaining clinical approval. The overall benefits, however, are limited and are often associated with unwanted side effects [6,7]. Recent studies have shown that VEGFC and VEGFR3 expression is elevated in human colorectal cancer specimens, including on lymphatic endothelial cells (LEC) and macrophages [8]. While some of the multi-kinase TK inhibitors are also shown to impact lymphangiogenesis [9] there are no lymphagiogenesis-specific drugs approved in the clinic. Recently SAR131675, a VEGRF3 specific inhibitor was developed and has shown tumor inhibition and modulation of tumor associated myeloid derived suppressive cells (MDSC) in several cancer types including a mouse model (4T1) of mammary carcinoma [10,11].

In this study we examine the ability of SAR131675, to control tumor growth in a murine CLM model and its effects in the modulation of both the adaptive and the innate arms of the immune response within the liver and tumor.

## 2. Materials and Methods

### 2.1. Animals and CR-LM Induction

All experiments were conducted in accordance with Austin Hospital Animal Ethics committee guidelines. Male CBA mice (ARC, Perth, Australia) aged 9–11 weeks and weighing between 20 and 25 g were used. Animals were kept in standard cages in a climate-controlled environment, exposed to a 12-h light/dark cycle, with irradiated food pellets and water supplied *ad libitum*. CLM were induced as previously described, utilizing an intrasplenic injection of an in vivo passaged primary colon carcinoma cell line (MoCR), which was originally derived from a dimethyl hydrazine (DMH)-induced primary colon carcinoma in the CBA mouse [12]. Briefly, 5 × 10^4^ tumor cell/50 µL suspension were slowly injected into the exteriorized spleen of anesthetized mice followed by splenectomy.

### 2.2. Treatment Protocol

Tumor induced mice were randomly assigned into control and treatment groups (7 mice per group). Control groups received 15 mL/kg drug vehicle (0.6% methylcellulose/0.5 Tween 80) and treatment groups received SAR131675 at 120 mg/kg body weight suspended in the vehicle solution, by oral gavage from 1 day post tumor induction until the endpoint. At each endpoint (days 10, 16 and 22 post tumor induction), a group of 7 control and 7 treatment mice were terminally anesthetized, and laparotomy was performed. Following heart puncture, the liver was slowly perfused with 10 mL saline solution via the portal vein, excised and harvested. The median and left lateral liver lobes were separated and placed in formalin (10%) (Sigma Aldrich, Castle Hill, NSW, Australia) for 24 h and then transferred in 70% ethanol until further processing. When tumors were visible, tumors in the right superior lobes were macroscopically dissected, separating tumor from the liver parenchyma and placed separately into digestion media on ice for fresh live cell analysis by flow cytometry.

### 2.3. Tumor Burden and IHC Assessment

The formalin fixed livers were used to determine the tumor burden. The livers were sliced into 1.5 mm sections using a multi-blade tissue fractionator and placed on a transparent plastic petri dish with 70% ethanol and photographed. The tumor area and liver area on the images were traced separately using the Image Pro plus software to calculate, whenever possible, the tumor burden = (tumor area)/(liver area) × 100.

Up to 5 non-consecutive sections per animal were selected and paraffin embedded for hematoxylin and eosin (H&E) cellular staining and IHC. The percentage of viable tumor as a proportion of whole tumor was calculated using H&E images and Image Pro plus software based on the formula: viable tumor area = (total area-necrotic area)/(total area) × 100. IHC was used to quantify the expression of CD34, podoplanin, Ki-67, F4/80 and VEGFR-3. Antibody source and dilution were as outlined in Appendix A and negative controls were stained without the corresponding primary antibodies. Positivity was calculated using the formula positivity = (number of positive pixels)/(total number of pixels) × 100.

### 2.4. Image Caption and Analysis Algorithm

Slides were scanned at 20× magnification (Aperio Scanscope AT Turbo; Leica Biosystems Wetzlar, Germany using Perio eSlide Manager software version 12.3.3.5049, Leica Biosystems Wetzlar, Germany). Image analysis software (Aperio ImageScope version 12.3.2.8013, Leica Biosystems Wetzlar, Germany) was used to calibrate and measure the degree of staining within designated areas in the images. An algorithm was set for each antibody in order to minimize the detection of background positive staining. These measurements were used to calculate a positivity score using the formula positivity = (number of positive pixels)/(total number of pixels).

### 2.5. Flow Cytometry

At endpoints of 10-, 16- and 22-days following tumor induction, livers were excised. While day 10 samples were processed whole, day 16 and 22 samples were macroscopically dissected into tumor and tumor free liver, which were then processed into single cell suspensions for analysis by flow cytometry according to a modified established protocol [13]. Tissue samples were incubated in digest medium (collagenase II 1 mg/mL, collagenase IV 1 mg/mL, DNase 25 U/mL in DMEM) for 40 min at 37 degrees. Samples were then passed through a 70 micron filter and treated with red blood cell lysis buffer. Cells were adjusted to a concentration of 1 × 10^7^/mL in ice-cold FACS buffer and incubated with 1 µL/mL of FC blocker (anti-CD32/CD16, BD Biosciences) for 10 min. Cells were then incubated with antibody cocktails to identify either lymphocyte populations (CD45-BV510, CD3-PE, CD8-PECy7, CD279(PD-1)-APC, CD4-APC-Cy7 (BD Bioscience) and CD103-FITC (Miltenyi biotech)) or myeloid cell populations (CD45-BV510, Ly6C-PE, CD274 (PD-L1)-PECy7, CD11b-APC-Cy7 (BD Bioscience) and F4/80-FITC (Miltenyi biotech). Details for antibody cocktails are found in Appendix A. The viability dye, DAPI (4′,6-diamidino-2-phenylindole, Sigma-Aldrich) was used to gate out dead cells. Single color and fluorescence minus one control were used to optimize and compensate the antibody panel. Samples were analyzed on a FACS CANTO II (BD Biosciences) and data analyzed by FlowJo^TM^ software version 10.7.1 (BD Biosystems, Ashland, OR, USA).

### 2.6. Statistical Analysis

Data were statically analyzed using either parametric or non-parametric tests as appropriate using Graphpad Prism 8.4 (GraphPad Software, Inc., San Diego, CA, USA). Data is presented either as the mean value of each group ± standard error of the mean or as box plots. Box plot data represented as the median percentage for each group as shown by the middle bar, the upper and lower box limits refer to 75th (Q3) and 25th (Q1) percentiles and the difference between the two is equal to the inter quartile range (IQR), while the upper and lower whiskers equal Q3 + 1.5 × IQR and Q1 − 1.5 × IQR respectively. All statistical tests were two-sided and a *p* value of <0.05 was considered statistically significant. Pearson correlation coefficient matrix were performed to determine relationships between phenotypic populations identified by flow cytometry.

## 3. Results

### 3.1. CRLM Growth Kinetics Following SAR131675 Treatment

Livers were collected on day 10, day 16 and day 22 following tumor induction. Macroscopic examination of livers at day 22 showed large and prolific tumors in the control group, while there was a visible dramatic reduction in tumor presence in the SAR131675 treatment group (Figure 1A).

Tumor burden analysis also demonstrated significant reduction in tumor presence at day 22 in the SAR131675 treated group, which equated to an 84% inhibition of tumor growth when compared to the control group (*p* < 0.0001, Figure 1B). To investigate if the treatment had any effect on the seeding of the tumor, the number of tumor nodules were counted per liver area on day 10 and day 16 time points using the H&E images, while Day 22 tumors became coalescent and not able to be accurately counted. The number of tumors present per liver area did not significantly differ between control and treatment groups at both day 10 (*p* = 0.92) and day 16 time points (*p* = 0.93, Appendix A) suggesting that SAR131675 treatment did not affect tumor seeding.

It was observed that SAR131675 treatment did not significantly affect the % tumor viability, despite the significant inhibition of tumor growth (Figure 1C, Appendix A).

Proliferation of tumor cells was assessed using Ki-67 staining, a general marker expressed in all stages of mitosis. Treatment with SAR131675 did not significantly alter the percentage of proliferating cells at each time point compared to control (Figure 1D, Appendix A). While Ki67 was used to identify cells in the process of mitosis, the time it takes for a cell to complete the process can vary, therefore this staining is not an accurate evaluation of the proliferation rate. We also assessed apoptosis by caspase-3 staining and found that there was no significant difference between the groups (*p* = 0.91).

### 3.2. SAR131675 Treatment Did Not Affect Tumor Vascularization and Lymphatic Density

The effect of SAR131675 on tumor vascularization, (angiogenesis and lymphangiogenesis) was studied using CD34 (progenitor endothelial cell marker) and podoplanin (a LEC marker). Angiogenesis was present by day 10 and significantly increased by day 16 (Control group *p* = 0.045, Figure 2A and Appendix A), confirming our previous findings [14]. SAR131675 treatment did not significantly change the percentage of CD34^+^ staining in the treated tumors compared with the control group at any time point, unlike the blood vessel reduction and distinctive pattern of necrosis observed when the angiogenesis inhibitor sunitinib was used in a previous study [15]. Similarly, the percentage of podoplanin staining did not significantly change at each of the three timepoints between control and SAR131675 treated groups, (Figure 2B, Appendix A).

### 3.3. Macrophages Are Reduced within the Liver and CRLM by SAR131675 Treatment

Macrophages were identified by F4/80 staining. In the tumor, macrophages primarily accumulate peritumorally and around vessels (Figure 3A, Appendix A), while in the liver they are found dispersed throughout the liver parenchyma (Figure 3C). SAR131675 treatment did not significantly alter macrophage infiltration within the tumor environment. Within the liver parenchyma, however, SAR131675 treatment significantly reduced the F4/80 positivity only at day 22 (*p* = 0.018; Figure 3D).

### 3.4. SAR131675 Expression of VEGFR3 in Tumor and Liver Parenchyma

VEGFR3 staining indicates that it is expressed on vascular structures, tumor cells and infiltrating immune cells. Tumors were highly VEGFR-3 positive on days 10 and 16, while the positivity significantly reduced on day 22 in both control and treated groups compared with their respective groups at day 16. However, SAR131675 treatment did not significantly affect VEGFR3 expression when compared to the controls at each time point in either tumor or liver parenchyma (Figure 4B,D and Appendix A). At the relevant time point of Day 22, where tumor burden was significantly reduced, the expression of VEGFR3 was not altered by SAR131675 treatment. This finding is not surprising since SAR131675 treatment inhibits the activity of VEGFR3 rather than its expression.

### 3.5. SAR131675 Treatment Significantly Alters the Balance of Lymphocytes and Myeloid Cells

Changes in the proportion of different leukocyte subpopulations within the total CD45^+^ leukocyte pool in tumor tissue and in liver parenchyma were studied by flow cytometry (Figure 5A). We noticed that in liver samples, the distribution of leukocytes (CD45^+^ cells) was altered within the different size gates that are established for small lymphocytes, myeloid cells and a combined gate (to capture both lymphocytes and myeloid cells) in SAR131675 treated sample at day 22, compared to control samples (Figure 5B) [16]. While leukocytes were reduced across all the cell size gates, the reduction was significant within the myeloid and combined groups. The distribution of CD45^+^ leukocytes in liver or tumor were not affected by SAR131675 treatment at day 10 and day 16 (data not shown) and at day 22 in the tumor (Figure 5B).

The accumulation of leukocyte populations has been shown to occur in tumor and adjacent liver parenchyma by us and others [17,18] and in this study we compared myeloid (CD11b^+^) and lymphocyte (CD3^+^) populations present in these tissues within the combined size gate (Figure 5C). The results demonstrate that the proportion of myeloid cells in liver increase between day 16 and 22 within the control group (Figure 5C) confirming previous reports (as above). SAR131675 treatment, however, prevented this accumulation within the liver parenchyma, resulting in significantly reduced myeloid cell levels at day 22, (Figure 5C). In contrast the proportion of CD3^+^ lymphocytes, significantly increases with SAR131675 treatment at day 22 within the liver, compared with the control group (Figure 5C). Similar changes occurred in the SAR131675 treated tumor tissues with a reduction in myeloid cells and increasing CD3^+^ lymphocytes at day 22, however the magnitude of change is less pronounced (Figure 5C).

### 3.6. SAR131675 Treatment Differentially Modulates the Infiltration of Myeloid Subsets

The myeloid cells within the leukocyte population in solid tissues comprises a heterogeneous pool of populations with differential expression of Ly6C that identified monocytic (Ly6C^high^) and granulocytic (Ly6C^low^) subsets as shown in Figure 6A. These may be of a mature phenotype, identified as macrophages by the expression of marker F4/80 (F4/80^+^) or as immature F4/80 negative cells (F4/80^−^), including the highly immunosuppressive cancer associated MDSC (Appendix A). The SAR131675 treatment not only reduced the overall proportion of myeloid cells (Figure 5C) but analysis of the myeloid cell composition shows that SAR131675 treatment differentially regulated both mature and immature myeloid subsets. Thus, the granulocytic macrophage population (F4/80^+^ Ly6C^low^), which comprises the largest macrophage subset in both liver and tumor tissues, was significantly increased by SAR131675 treatment in the liver, while significantly decreased in the tumor tissues (Figure 6B). In contrast, the monocytic derived macrophage population representing a smaller proportion of the total macrophages in both liver and tumor tissues, did not significantly alter in proportion with the SAR131675 treatment in either tissue (Figure 6B).

One way myeloid subsets are known to modulate immune responses is by altering the surface expression of the checkpoint ligand PD-L1 [19]. We investigated whether SAR131675 treatment changed the PD-L1 expression on CLM macrophages (Appendix A, gating strategy for myeloid cell phenotype). We found the proportion of granulocytic PD-L1^+^ macrophages significantly increased in the liver, while they reduced in the tumor, albeit not significantly (Figure 6D) in line with the overall granulocytic macrophage changes as described above. In the tumor, the proportion of PD-L1^−^ monocytic macrophages significantly increased with the SAR131675 treatment, albeit against low control levels, again reflecting changes in the overall monocytic macrophage proportion. In general, most of the cells of both granulocytic and monocytic subsets are PD-L1 positive and the SAR131675 treatment did not alter their PD-L1 expression. Together these results indicate that the reduction in the granulocytic macrophages, the largest myeloid population in the tumor, may be particularly significant in the context of tumor control.

The largest proportion among the immature MDSC subtypes in the liver and tumor tissue were the granulocytic MDSC (F4/80^−^ Ly6C^low^), and SAR131675 treatment significantly reduced this subset in the liver. In the tumor, SAR131675 treatment significantly increased the monocytic MDSC (F4/80^−^ Ly6C^high^) subset (Figure 6C).

Granulocytic MDSC expressing checkpoint marker PD-L1^+^ made up the largest phenotypic population of MDSC in the liver. Both PD-L1^+^ and PD-L1^−^ subsets were reduced by SAR131675 treatment. These changes also reflected the changes in the proportions of the total MDSC subsets, thus the SAR131675 treatment did not alter their PD-L1 expression. In the tumor however, the proportion of PD-L1^+^ monocytic MDSC population significantly increased (Figure 6E).

Together these results indicate that the alterations of myeloid cell subpopulations by SAR131675 treatment, may be particularly significant in the context of tumor control, possibly through their regulation of lymphocytes.

### 3.7. SAR131675 Treatment Modulates the T Lymphocyte Responses to CRLM

We have demonstrated earlier (Figure 5) that SAR131675 treatment also modulates the adaptive arm of the immune response to CRLM. The SAR131675 treatment effect was further assessed by examining the changes in CD3^+^ T cell subtypes; CD4^+^ helper, CD8^+^ cytotoxic and CD4^−^ CD8^−^ double negative (DN) T cells as a proportion of total CD45^+^ leukocytes (gating strategy for lymphocyte cell phenotype Appendix A).

While the overall CD3^+^ T cell proportion of leukocytes increased following SAR131675 treatment at day 22 (Figure 5C), further analysis showed that only the CD8^+^ T cell subset significantly increased in the liver, while in the tumor, SAR131675 treatment significantly increased the proportion of the CD4^+^ T cell subset (Figure 7A). The expression of PD-1 checkpoint receptor also significantly increased on CD8^+^ T cells in the liver and on both CD4^+^ and CD8^+^ T cells in the tumor suggesting recent activation of these cells. (Figure 7B).

CD103 is one of the markers used to identify cells that differentiate and become tissue resident T cells (T_Res_) which no longer recirculate into the blood. CD103 has been shown to bind to E cadherin and this may be one of the mechanisms by which T_Res_ are retained within peripheral tissues. Indeed, we have shown previously that MoCR tumors are strongly positive for E cadherin [20]. T_Res_ CD8^+^ T cells have been previously associated with improved survival of CRLM patients [21,22] and thus were examined here in both liver and tumor. SAR131675 treatment increased the proportion of CD103^+^ CD8^+^ T cells in the liver, however the percentage of CD8^+^ T_Res_ expressing PD1 did not increase (Figure 7C–E).

### 3.8. Ratios of Myeloid Subsets/T Cell Subsets Are Significantly Altered by SAR131765 Treatment

Clinical and experimental studies indicate that the both the absolute numbers and the ratios of certain leukocyte subsets within the TME or in blood circulation could be predictive indicators of patient survival prospects. Infiltration of high numbers of lymphocytes, with the exception of regulatory T cells and helper Th2 cells, are associated with better outcomes [23]. In contrast, infiltration of myeloid cells is associated with worse prognosis [24], as is a high neutrophil to lymphocyte ratio [9]. Additionally, changes in these ratios by the cancer treatment may predict the treatment efficacy [25,26].

A comparison of the ratios of CD11b^+^ myeloid cells to CD3^+^ T cells indicated that SAR131675 treatment significantly decreased this ratio in both liver and tumor (Table 1), indicating a shift in the balance of immune response, favoring an anti-tumor effect.

The ratio of different immune cell subsets indicates the relative frequencies to each other within a tissue. In this study SAR131675 treatment significantly altered the relative frequencies of CD8^+^ and CD4^+^ T cell subsets (Table 2). Within SAR131675 treated liver tissues, the increase in CD8^+^:CD4^+^ T cell subset ratios indicated that CD8^+^ T cell frequencies increased relatively to CD4^+^ T cells and DNT cells (Appendix A), while activated PD1^+^ CD8^+^ T cells increased significantly relative to PD1^+^ CD4^+^, but not PD1^+^ DNT cells. In the tumor, however, SAR131675 treatment decreased CD8^+^ T cells’ frequency compared to CD4 and to a lesser extent DNT (Table 2, Appendix A), while there was no effect on the relative frequency of activated PD1^+^ T cell subsets.

Literature suggests that the ratio of granulocytic to monocytic MDSC and macrophages in tumors can impact survival [27,28]. Granulocytic MDSCs preferentially accumulate in the TME, the periphery and blood, in several types of cancer [29]. We examined changes effected by the SAR131675 treatment in these ratios for both PD-L1^+^ and PD-L1^−^ phenotypes in the liver and tumor tissues. This ratio was significantly reduced in the liver MDSC PD-L1^+^ subtypes. While in the tumor the ratio for both macrophage and MDSC PD-L1^+^ and PDL1^−^ subsets were reduced (Table 3). This indicates that SAR131675 treatment reduces the frequency of granulocytic MDSC PD-L1^+^ in the liver, and the frequency of both granulocytic macrophages and MDSCs compared to monocytic counterparts in the tumor and that this may play an important role in the suppression of tumor growth.

#### Correlation Results

Pearson correlation coefficients were determined for all leukocyte subtypes examined in this study in the control liver and tumor tissues to reveal any significant associations between different subtypes. We then determined these coefficients for the same leukocytes SAR131675 treated groups (Appendix A). We focused on the association of relative frequencies of myeloid populations with that of T cell phenotypes; T_Res_ (CD8^+^ CD103^+^) and CD4^+^ to determine if there were relationships that may impact on these T cells.

Within the livers of the control group, there was a significant positive correlation between CD8^+^ T_Res_ and PD-L1^+^ monocytic MDSC, while there was a negative association between CD8^+^ and CD8^+^ T_Res_ with PD-L1^−^ granulocytic MDSC. Interestingly, in the SAR131675 treated livers, these associations were ablated (Table 4, Appendix A).

In the tumor, CD8^+^ T cells and CD8^+^ T_Res_ positively correlate with monocytic MDSCs, regardless of PD-L1 status, while SAR131675 treatment ablated this correlation (Table 4 and Appendix A). Interestingly, SAR131675 treatment showed a negative correlation between PD-L1^−^ granulocytic macrophages and CD8^+^ T_Res_ in the tumor.

Pearson correlations of myeloid populations and CD4^+^ T cells indicated that within the liver there were significant negative associations between PD-L1^−^ macrophage phenotypes and that SAR131675 treatment ablated these (Table 5, Appendix A). Interestingly, SAR131675 treatment resulted in a positive association between PD1^+^ CD4^+^ T cells and PD-L1^−^ monocytic MDSC (Appendix A).

In the tumor, CD4^+^ and PD1^+^ CD4^+^ T cells positively associated with monocytic MDSCs, regardless of PD-L1 status, which was not changed by SAR131675 treatment (Table 5 and Appendix A). PD-L1^+^ monocytic tumor macrophages in the control group also showed a positive association between CD4^+^ and PD1^+^ CD4^+^ T cells which was ablated by SAR131675 treatment (Table 5 and Appendix A).

Surprisingly, these results indicated the populations of myeloid cells that may be influencing the presence of T cell subsets in the tissue were not the most predominant myeloid subpopulation within the liver, PD-L1^+^ granulocytic MDSC. It is clear that the negative association or influence of myeloid cells on CD8^+^ or CD4^+^ T cell subsets are mostly eliminated by SAR131675 treatment in the liver. While in the tumor positive associations between monocytic MDSC cells and CD4^+^ T cells are retained in both treatment groups.

Taken together with the relative changes in leukocyte subsets in liver and tumor these associations indicate that SAR131675 treatment changes the way the myeloid cells associate with T cell populations. Overall, the presence of myeloid cells is greatly reduced in the liver of SAR13175 treated mice. The remaining myeloid cells in SAR131675 treated animals no longer have the associations they demonstrate in the control group; macrophages no longer have significant negative associations with CD8^+^ T_Res_ and CD4^+^ T cells in the liver. In the tumor, however, the overall presence of myeloid cells is unchanged, but the proportions of granulocytic PDL1^−^ macrophage population reduced by SAR131675 appear to have a significant negative association with CD8^+^ T_Res_. Whereas, increasing monocytic MDSC subsets in the tumor retained a positive association with CD4^+^ T cells. Possibly reflecting the diminishing suppressive influence of the granulocytic myeloid cells while changes in the monocytic populations could support effective anti-tumor responses.

## 4. Discussion

The VEGFR3 specific inhibitor SAR131675 was shown in other studies to reduce tumor growth and metastasis in a number of mouse cancer models including breast, colorectal, prostate and pancreatic cancers and found to be well tolerated [10]. The mechanisms proposed for the SAR131675 effects are inhibition of lymphangiogenesis, angiogenesis and selective inhibition of myeloid cells including TAM and MDSCs [10,11].

In this study we investigated the effects SAR131675 treatment had in a CRC mouse model of liver metastasis. In accord with the earlier studies, we observed significant tumor inhibition. However, unlike the earlier studies [10] we did not observe any reduction in angiogenic or lymphangiogenic staining density, nor changes in tumor morphology or in the density of proliferating cells. In a previous study using the angiogenesis inhibitor Sunitinib [15] we observed distinctive morphology including clear reduction in tumor vessels and tumor burden in the treated animals. SAR131675 treatment, however, did not show such changes, suggesting that angiogenesis does not play a significant role in the tumor reduction. These results suggest there may be some differences in the mechanisms leading to tumor inhibition, not involving angiogenesis or lymphangiogenesis. In our model the tumor is VEGFR3 positive in contrast to tumors used in the other studies [8,10]. Therefore, it is likely a significant component of the tumor inhibition is through direct inhibition of VEGFR3 induced tumor growth. VEGFR3 expression was reported in many types of human cancers including colorectal and gastric cancers, which often co express VEGFC giving opportunity for autocrine regulation of signaling and growth promotion [30,31]. While the proportion of staining of lymphangiogenic, angiogenic and proliferation markers is not significantly different between control and treated tumors, the significantly reduced tumor volume suggests that tumor cells are taking longer to complete the process of mitosis.

Immunohistochemical staining revealed significant reduction in macrophage infiltration within the liver parenchyma, however unlike the Alam study [10] we did not observe reduction in TAMs within the tumor. This discordance could be due to different types of tumors used. It is now well established that in the presence of tumors, macrophages, and immature myeloid cells increase in circulation and accumulate within the tumor, adjacent parenchyma and spleen [11]. Immature myeloid cells depending on local signals can differentiate into macrophages, neutrophils and even became endothelial cells [32]. High levels of macrophages and immature myeloid cells which are also referred to as MDSCs are predictive of worse prognosis and support tumor growth in many ways including creating an immunosuppressive environment, inhibiting activation, proliferation and migration of T cells and contributing to vascularization [33,34].

In this study we demonstrated that SAR131675 treatment significantly reduced the total myeloid cell population within the liver and to a lesser extent in the tumor of mice with CRLM. Interestingly the different subsets of macrophages (F4/80^+^) and MDSCs (F4/80^−^) were modulated differentially within each tissue. The proportion of granulocytic macrophages increased in the liver with SAR131675 treatment, while the proportion of granulocytic MDSCs decreased, suggesting that a proportion of these may have been matured into granulocytic macrophages by the treatment. In contrast the proportion of granulocytic macrophages significantly decreased in the tumor, while there was a significant increase in the monocytic fraction of MDSCs. The significance of these changes and how they contribute to tumor inhibition are not yet clear and need to be further investigated. Differential modulation of myeloid subtypes with SAR131675 treatment within the tumor tissues were also reported by Espagnolle et al. in a mouse breast cancer model [11]. Although their characterization of myeloid cell subtypes was different to ours (as they used a LY6G staining to differentiate granulocytic from monocytic myeloid subtypes), they also reported a significant decrease in granulocytic macrophages. Moreover, while they reported an increase in monocytic macrophages, we observed an increase in monocytic MDSCs. Espagnolle et al., reported that the monocytic macrophages had an immunostimulatory potential, while their granulocytic macrophages and MDSCs inhibited T-cell stimulation in in vitro assays [11].

Our study additionally, demonstrated significant changes effected in T cell levels and subtype composition by SAR131675 treatment. There was an overall increase in the total T cell population which was more pronounced within the liver parenchyma. Similar to the myeloid cell subtypes, T cell subtypes were differentially regulated, thus CD8^+^ T cells which are usually associated with tumor cytotoxic properties [35], were increased within the liver parenchyma as a proportion of the total T cell population, while the proportion of CD4^+^ T cells significantly increased within the tumor. This was rather unexpected since the majority of published studies link CD8^+^ T cells with effector tumor killing, while helper or regulatory functions are usually associated with CD4^+^ T cells [35,36]. More recent studies, however, demonstrate the importance of CD4^+^ T cells in anti-tumor responses, with a number of clinical and experimental studies reporting cytotoxic CD4^+^ T cells recognizing tumor neoantigens and or self-antigens [37,38,39]. Spitzer et al. reported that the transfer of CD4^+^ T cells isolated from animals with effective tumor therapy provided longer lasting antitumor protection to untreated recipients than the transfer of CD8^+^ T cells from the same donors. They also identified a similar CD4^+^ T cell subtype in melanoma patients responding to immunotherapy [37,38,39]. Garcia-Martinez et al. also reported a high CD4^+^/CD8^+^ ratio was a predictor of increased survival in breast cancer patients undergoing chemotherapy [25,26]. More recently Oh et al. demonstrated the existence of cytotoxic CD4^+^ T cells in bladder tumors that can kill autologous tumors in an MHC class II-dependent fashion upon anti-PD-L1 treatment [40]. With this background our finding that VEGFR3 inhibition induces upregulation of CD4^+^ T cells could indicate an important role in inhibiting tumor growth.

Taken together, all the changes we observed, namely reduction in total macrophages and MDSCs, increase in CD3^+^ T cells, changes in the myeloid and T cell subsets and their ratios towards an anti-tumor responsive state, indicate that the modulation of the immune response through reduction in MDSC is a major mechanism of the VEGFR3 inhibitor in restraining tumor growth. The exact mechanism of how this occurs however is not yet clear and it may vary among tumors. The inhibitor could act directly on the fraction of myeloid cells expressing VEGFR3 [11,41] inducing their death or preventing their tumor infiltration or ablating their immunosuppressive functions. Tacconni et al. reported systemic administration of VEGFC increased TAM density in mouse primary CRC tumor models, while treatment with mF431C1, an anti-VEGFR3 antibody, significantly reduced their density. Additionally VEGFR3 activation resulted in immunosuppressive TAMs, and in contrast the mF431C1 treatment resulted in TAMs expressing genes involved in a productive immune response [8]. VEGFR3 is also expressed on tumor lymphocytic vessels and when VEGFC, often secreted by the tumor, binds to it, LECs produce chemokines that attract TAMS into the tumor environment. These, in turn, produce additional VEGFC that acts in a positive feedback loop to maintain the active state of LECs. Activated LECs also directly inhibit T cell activation [8] and thus act in concert with TAMS/MDSCs to maintain an immunosuppressive tumor environment. Activation of VEGFR3 on tumor cells, also induces the production of chemo attractants for TAMs and MDSCs, further enhancing immunosuppression.

Changing the tumor microenvironment from immunosuppressive to immunostimulatory holds the biggest promise for more effective tumor therapies as is attested with recent successes of the anti-PD-L1 treatment, which however met with limited success in CRC patients. Success of PD-1/PD-L1 inhibition in cancer treatments depends on tumor mutational load and high T cell infiltration [42]. Additionally, PD-L1 tumor positivity is used as an indicator for responsive patients to PD-1/PD-L1 inhibition [43]. High levels of PD-L1 expression on macrophages were shown to be associated with longer OS in anti-PD-1 treated patients [19]. Our study shows that while there was significant reduction in the overall myeloid cell population and changes in the proportions of specific subsets, their PD-L1 expression, which in general was high, did not significantly change. Taken together with the increased CD3^+^ T cell infiltration in response to SAR131675 treatment, the results suggest that a combination of SAR131675 treatment with PD-1/PD-L1 inhibition could increase the anti-tumor efficacy.

There are some limitations in this study, one of them being that the model requires the removal of the spleen to avoid the establishment of spleen bed tumors. We believe that this does detract from the results. In some other publications splenectomy is associated with less tumor growth as the spleen presents a reservoir of immunosuppressive myeloid derived cells, both macrophages and MDSCs [44,45]. Indeed, in another study using this model with hemi-splenectomy, it was observed that CRCLM growth was more vigorous [46]. Another limitation is that we used different lobes of the liver to determine tumor burden, and for flow cytometry, while we were consistent using the same lobe from controls and treated animals for each procedure, it would have been ideal to use the whole liver for each test. Lastly our study is an exploratory study that has identified significant changes in immune cell composition associated with tumor inhibition, further characterization of specific immune cell subgroups either by in vitro assays or gene expression analysis would shed light on their role in tumor inhibition. Additionally, the use of VEGFR3 deleted tumor cells and/or VEGF-C inhibitors could further elucidate the mechanisms of SAR131675 tumor inhibition.

## 5. Conclusions

Taken together our study demonstrates that the VEGFR3 specific inhibitor SAR131675 is effective and well tolerated by the animals. It rebalances the TME towards an antitumor immune response by reducing the immunosuppressive myeloid derived cells and increasing the tumor fighting CD3^+^ T cells and, in particular, CD4^+^ T cells which in other studies is an indicator of an effective therapy. The results also suggest that VEGFR3 specific inhibition may be used in combination with other treatments including tumor resection and/or immunotherapies and may be a suitable treatment for CRCLM patients following resection.

## Figures and Tables

**Figure 1 cancers-14-02715-f001:**
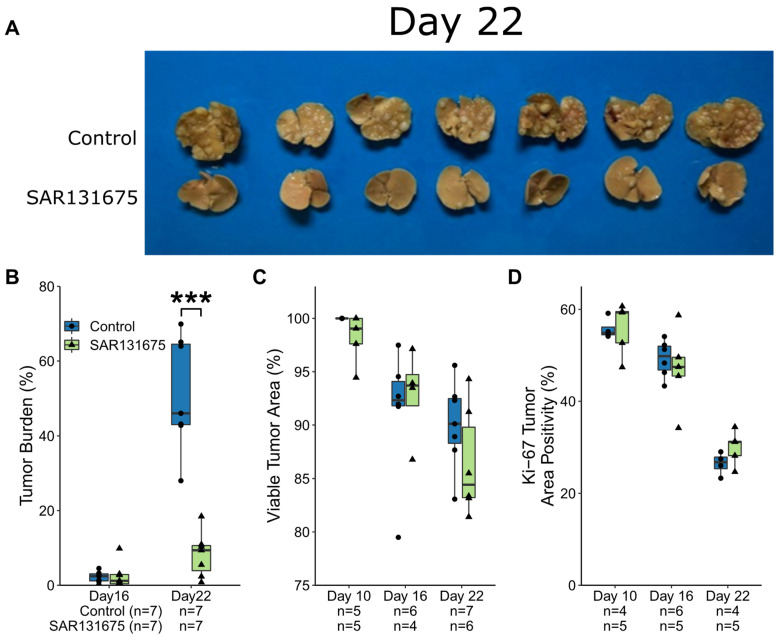
SAR13675 dramatically reduced CRLM tumor burden. (**A**) Images of livers collected on Day 22. (**B**) Tumor burden as a percentage of total liver area. (**C**). Viable tumor area as a percentage of total tumor area. (**D**) Ki67 positive tumor area as a percentage of total tumor area. *** *p* < 0.001.

**Figure 2 cancers-14-02715-f002:**
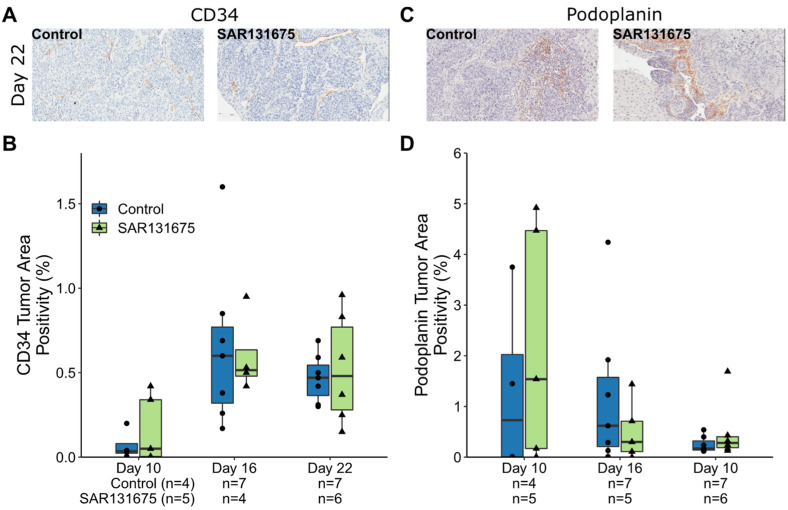
Effects of SAR131675 on tumor vascular density. (**A**,**C**) Representative IHC images of CD34 and podoplanin staining (brown color) in control and SAR131675 treated tumor sections. (**B**,**D**) Enumeration of CD34 and podoplanin positively staining expressed as a percentage of total tumor area.

**Figure 3 cancers-14-02715-f003:**
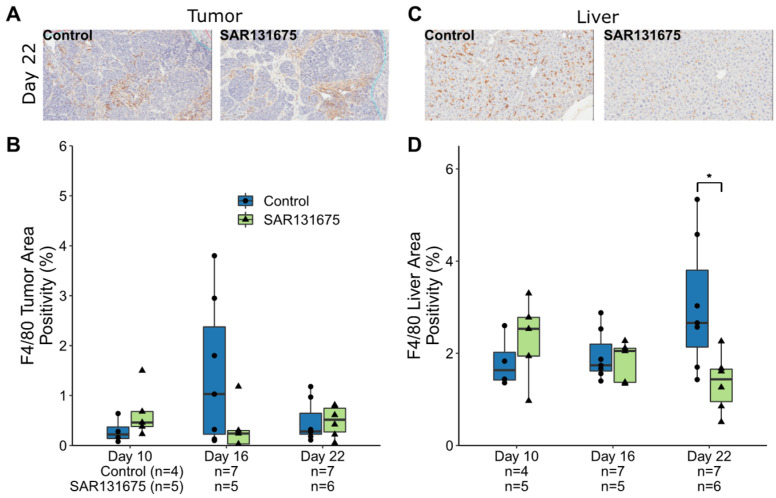
SAR13675 reduces F4/80^+^ cells in liver parenchyma. Representative IHC images of F4/80 staining cells (brown staining) in control and SAR13675 treated tumor (**A**) and liver (**C**) sections. (**B**) Enumeration of F4/80 positively staining in the tumor as a percentage of total tumor area. (**D**) Enumeration of F4/80 positively staining in liver as a percentage of total liver area. Significance is indicated by * while non-significance is unmarked (* *p* < 0.05).

**Figure 4 cancers-14-02715-f004:**
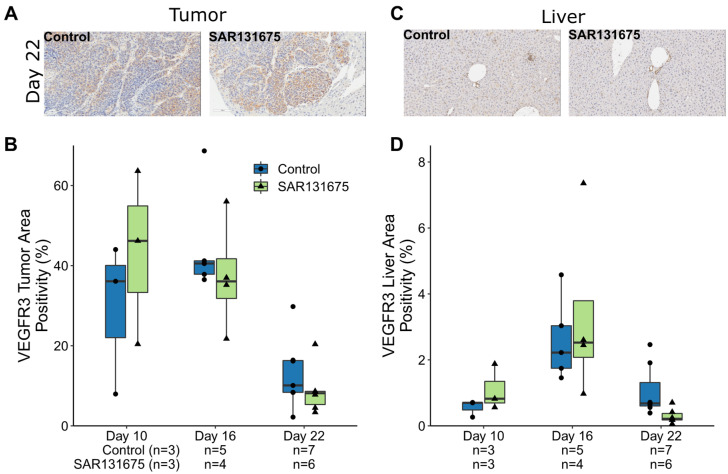
VEGFR3 staining in tumor and liver parenchyma. Representative IHC images of VEGFR3 staining cells (brown staining) in control and SAR13675 treated tumor (**A**) and liver (**C**) sections. (**B**) Enumeration of VEGFR3 positively staining in the tumor as a percentage of total tumor area. (**D**) Enumeration of VEGFR3 positively staining in liver as a percentage of total liver area.

**Figure 5 cancers-14-02715-f005:**
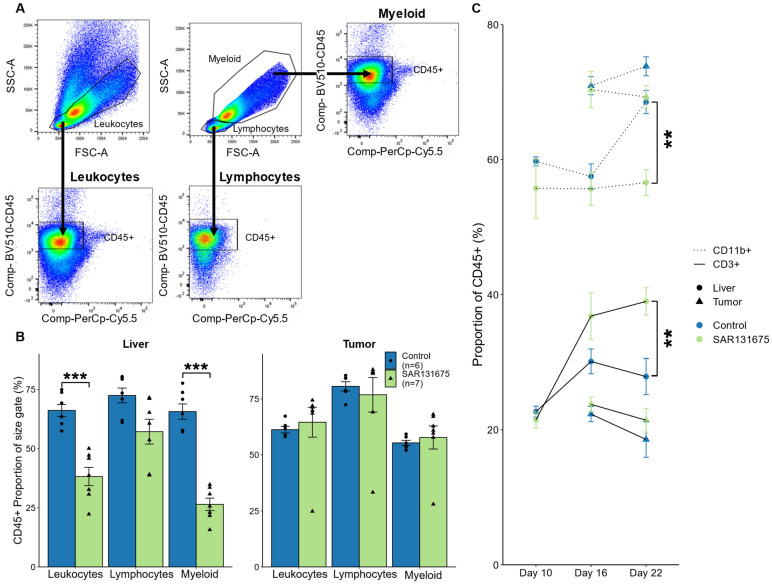
Percentage of CD3^+^ and CD11b^+^ cells as a proportion of CD45^+^ leukocytes in liver and tumor. Following CRLM tumor induction and SAR131675 treatment, liver and tumor tissues were isolated and analyzed by flow cytometry at days 10, 16 and 22. (**A**) Live, single, non-auto fluorescent cells within size and granularity gates for small lymphocytes, myeloid cells and a combination of both (Leukocytes) were analyzed for CD45^+^ staining. (**B**) The percentage of CD45^+^ leukocytes as a proportion of all live cells within the selected size gates on day 22. (**C**) The percentage of CD3^+^ and CD11b^+^ cells as a proportion of CD45^+^ cells in the combined leukocyte size gate at each time point for liver and tumor tissues. ** *p* < 0.01 and *** *p* < 0.001.

**Figure 6 cancers-14-02715-f006:**
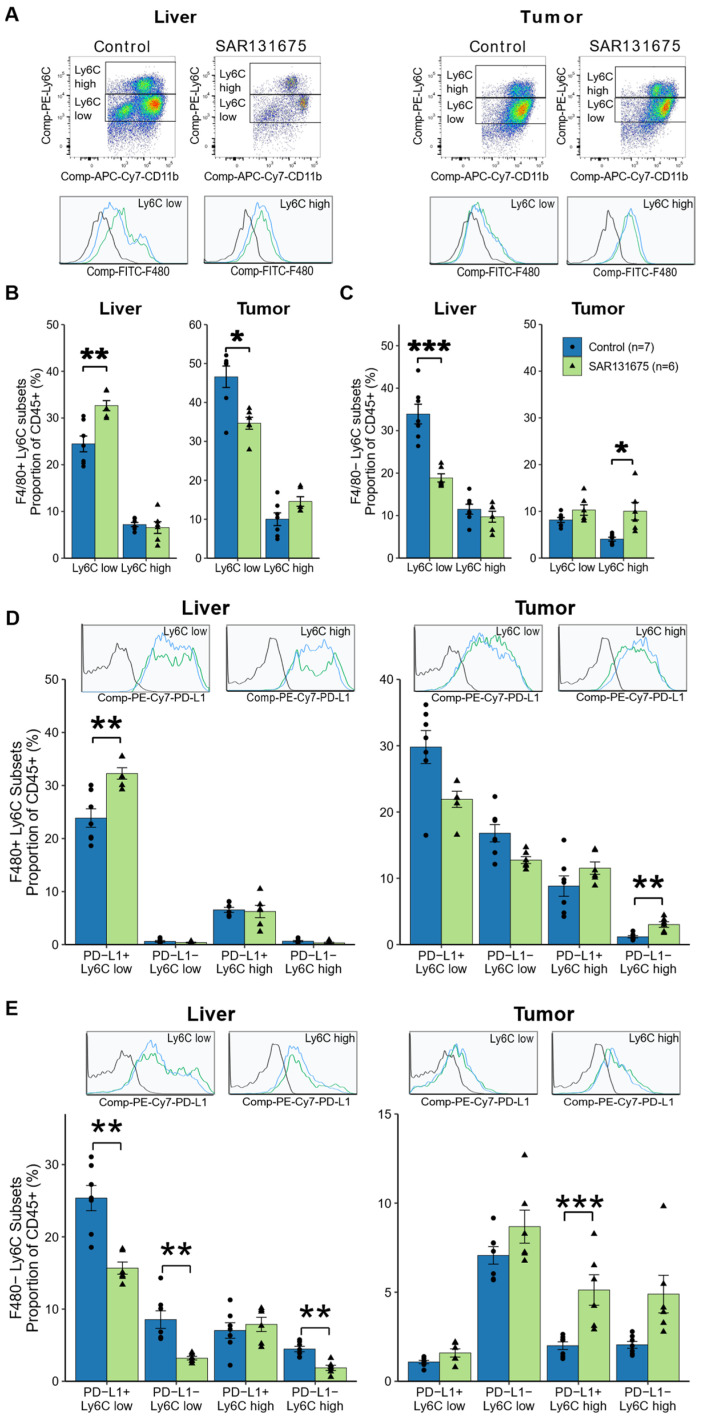
SAR131675 treatment alters the phenotype of myeloid cells in liver and tumor at day 22. Myeloid cells were defined as MDSC (F4/80^−^) or macrophage (F4/80^+^) while high and low expression on these subsets of Ly6C defined these as monocytic and granulocytic, respectively. (**A**) Representative dot plots of CD45^+^ CD11b^+^ cells and histograms of F4/80 fluorescence in Ly6C low and Ly6C high expression populations in liver and tumor. (**B**) Percentage of Ly6C^low^ and Ly6C^high^ macrophages and (**C**) MDSC. (**D**) Representative histograms of PD-L1 fluorescence in Ly6C low and high expression populations of F4/80^+^ macrophages and percentage of PD-L1^+^ and PD-L1^−^ in these populations as a proportion of CD45^+^ leukocytes (**E**) Representative histograms of PD-L1 fluorescence in Ly6C low and high expression populations of F4/80^−^ MDSC and percentage of PD-L1^+^ and PD-L1^−^ in these populations as a proportion of CD45^+^ leukocytes. Significance is indicated by * while non-significance is unmarked (* *p* < 0.05, ** *p* < 0.01 and *** *p* < 0.001).

**Figure 7 cancers-14-02715-f007:**
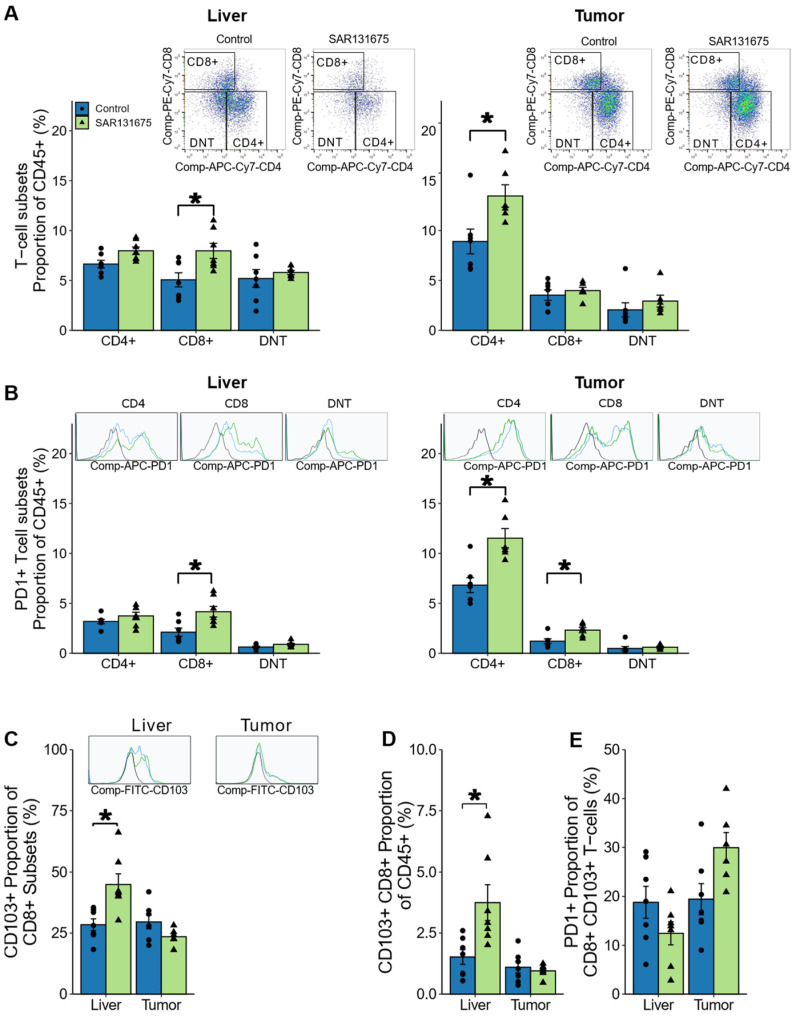
SAR131675 treatment alters T cells in liver and tumor at day 22. At day 22 CD3^+^ T cells were analyzed by flowcytometry for CD4^+^, CD8^+^ and double negative (CD4^−^ CD8^−^) subsets. (**A**) Representative dot plots of CD45^+^ CD3^+^ T cell subsets in liver and tumor and percentage of T cell subsets as a proportion of CD45^+^ leukocytes. (**B**) Representative histograms of PD1 florescence within the T cell subset and percentage PD1^+^ T cell subsets as a proportion of CD45^+^ leukocytes in liver and tumor. (**C**) Representative histograms of CD103 fluorescence for CD8^+^ T cells and percentage of CD103^+^ as a proportion of CD8^+^ T cells. (**D**) Percentage of CD103^+^ CD8^+^ T cells as a proportion of CD45^+^ leukocytes and (**E**) percentage of PD1^+^ cells as a proportion of CD103^+^ CD8^+^ T cells in liver and tumor. Significance is indicated by * while non-significance is unmarked (* *p* < 0.05).

**Table 1 cancers-14-02715-t001:** Ratio of CD11b^+^ Myeloid cells to CD3^+^ Lymphocytes in liver and tumor ^a^.

	**Liver**	**Tumor**
Control mean ± SD	4.73 ± 1.88	5.87 ± 1.94
SAR131675 mean ± SD	2.6 ± 0.40	3.09 ± 0.77
*p* value	**0.024**	**0.015**

^a^ Ratios are calculated as the relative ratio of total CD11b^+^ myeloid cells to CD3^+^ lymphocytes as a proportion of CD45^+^ leukocytes within the same size gate. Data represents average ± standard deviation. Significance calculated by Student *t* test. Significance *p* < 0.05 is indicated by bold text.

**Table 2 cancers-14-02715-t002:** Ratio of CD8^+^ to CD4^+^ T cell subsets in liver and tumor ^a^.

	**Liver**		**Tumor**	
**Subset**	**Total**	**PD1^+^**	**Total**	**PD1^+^**
Control mean ± SD	0.75 ± 0.20	0.65 ± 0.28	0.39 ± 0.08	0.18 ± 0.07
SAR131675 mean ± SD	1.01 ± 0.28	1.28 ± 0.82	0.30 ± 0.03	0.20 ± 0.04
*p* value	**0.071**	**0.094**	**0.0167**	0.6777

^a^ Ratios are calculated as the relative ratio of CD8^+^ to CD4^+^ CD3^+^ T cells of both total and PD1^+^ subsets within the CD45^+^ leukocyte population. Data represents average ± standard deviation. Significance calculated by Student *t* test. Significance *p* < 0.05 is indicated by bold text.

**Table 3 cancers-14-02715-t003:** Ratio of granulocytic (Ly6C^low^) to monocytic (Ly6C^high^) myeloid cells subsets in liver and tumor ^a^.

	**Liver**				**Tumor**			
	**Macrophage**		**MDSC**		**Macrophage**		**MDSC**	
	**PD-L1^+^**	**PD-L1^−^**	**PD-L1^+^**	**PD-L1^−^**	**PD-L1^+^**	**PD-L1^−^**	**PD-L1^+^**	**PD-L1^−^**
Control mean ± SD	3.69 ± 0.69	0.91 ± 0.28	4.23 ± 2.03	1.90 ± 0.47	3.81 ± 1.28	16.10 ± 6.78	0.57 ± 0.17	3.64 ± 1.11
SAR131675 mean ± SD	6.60 ± 4.23	1.73 ± 0.78	2.15 ± 0.69	2.12 ± 1.26	1.97 ± 0.48	4.65 ± 1.58	0.35 ± 0.15	2.09 ± 0.87
*p* value	0.15	**0.048**	**0.036**	0.70	**0.008**	**0.004**	**0.033**	**0.017**

^a^ Ratios are calculated as the relative ratio Ly6C^low^ to Ly6C^hi^ of Macrophage (F4/80^+^) and MDSC (F4/80^−^) CD11b^+^ cells of both PD-L1^+^ and PD-L1^−^ subsets. Data represents average ± standard deviation. Significance calculated by Student *t* test. Significance *p* < 0.05 is indicated by bold text.

**Table 4 cancers-14-02715-t004:** Associations for myeloid cell subsets correlations with CD103^+^ CD8^+^ T cell subset ^a^.

	Liver				Tumor			
Subset	Ly6C^lo^ PD-L1^+^	Ly6C^lo^ PD-L1^−^	Ly6C^hi^ PD-L1^+^	Ly6C^hi^ PD-L1^−^	Ly6C^lo^ PD-L1^+^	Ly6C^lo^ PD-L1^−^	Ly6C^hi^ PD-L1^+^	Ly6C^hi^ PD-L1^−^
MAC (F4/80^+^)								
Control	-	−0.849 *	-	-	-	-	0.931 **	-
SAR131675	-	-	-	-	-	−0.908 *	-	-
MDSC (F4/80^−^)								
Control	-	−0.835 *	0.825 *	-	-	-	0.839 *	0.845 *
SAR131675	-	-	-	-	-	-	-	-

^a^ Significant r values (* *p* < 0.05 and ** *p* < 0.01) over 0.7 between CD8 CD103^+^ T cells and myeloid cell populations calculated by Pearson correlation matrix.

**Table 5 cancers-14-02715-t005:** Associations for myeloid cell subsets correlations with CD4^+^ T cell subset ^a^.

	Liver				Tumor			
Subset	Ly6C^lo^ PD-L1^+^	Ly6C^lo^ PD-L1^−^	Ly6C^hi^ PD-L1^+^	Ly6C^hi^ PD-L1^−^	Ly6C^lo^ PD-L1^+^	Ly6C^lo^ PD-L1^−^	Ly6C^hi^ PD-L1^+^	Ly6C^hi^ PD-L1^−^
MAC (F4/80^+^)								
Control	-	−0.803 *	-	−0.755 *	-	-	0.962 ***	-
SAR131675	-	-	-	-	-	-	-	-
MDSC (F4/80^−^)								
Control	-	-	-	-	-	-	0.819 *	0.893 **
SAR131675	-	-	-	-	-	-	0.878 *	0.911 *

^a^ Significant r values (* *p* < 0.05, ** *p* < 0.01 and *** *p* < 0.001) over 0.7 between CD4^+^ T cells and myeloid cell populations calculated by Pearson correlation matrix.

## Data Availability

Data generated or analyzed during this study are included in this published article.

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
