# Peer review of "SAR131675, a VEGRF3 Inhibitor, Modulates the Immune Response and Reduces the Growth of Colorectal Cancer Liver Metastasis"

_cancers, 2022, doi:10.3390/cancers14112715_

Round 1

Reviewer 1 Report

In this article, Walsh et al investigated the effect of VEGFR3 inhibitor (SAR131675) on colorectal cancer liver metastasis. The results presented show significant reduction of tumor burden at later time point (day 22). Further analysis indicated a change in immune composition within tumor and surrounding liver. The data suggest an immunosuppressive environment upon inhibitor treatment. The results are well presented, and conclusions are justified. This paper has a therapeutic potential. However, it is not clear how VEGFR3 inhibitor is impacting the tumor burden and immune composition, which is also a major weakness. There are few other concerns about this study and need to be addressed.

  1. It is important to show TUNEL or other cell death staining on treated vs control group to supplement ki67 data.
  2. VEGFR3 expression was reduced at day22, the same timepoint showing reduced tumor burden upon SAR131675 treatment. Needs explanation.
  3. Is there a change in PDL1 expression on tumor cells upon SAR131675 treatment.
  4. It may be too much for this study to investigate mechanism of VEGFR3 inhibition. However, some experiments like injecting tumor cells upon VEGFR3 deletion or VEGF-C deletion will help.
  5. To rule out contribution of angiogenesis, the authors should include angiogenesis inhibitors like bevacizumab (avastin) or VEGFR2 inhibitors as a control in their experiment.

Author Response

In this article, Walsh et al investigated the effect of VEGFR3 inhibitor (SAR131675) on colorectal cancer liver metastasis. The results presented show significant reduction of tumor burden at later time point (day 22). Further analysis indicated a change in immune composition within tumor and surrounding liver. The data suggest an immunosuppressive environment upon inhibitor treatment. The results are well presented, and conclusions are justified. This paper has a therapeutic potential. However, it is not clear how VEGFR3 inhibitor is impacting the tumor burden and immune composition, which is also a major weakness. There are few other concerns about this study and need to be addressed.

We thank the reviewer for their assessment of our submission and their constructive comments and suggestions, addressing these will improve the quality of the paper.

  1. It is important to show TUNEL or other cell death staining on treated vs control group to supplement ki67 data.

We acknowledge the reviewer’s comment and agree with the reviewer that data on apoptotic cell death is a valuable for this study. We also used caspase 3 staining which stains apoptotic cells, however we did not find significant difference between control and experimental groups. We now include this information within section 3.1.

  1. VEGFR3 expression was reduced at day22, the same timepoint showing reduced tumor burden upon SAR131675 treatment. Needs explanation.

We acknowledge the reviewer’s comment, the reviewer is correct in their observation of reduced VEGFR3 expression at day 22. The reduction in VEGRF expression is seen in both control and treated groups and the difference in VEGFR3 expression between the groups is not significant. Day 22 is also the timepoint that difference in tumor burden is highly significant. This finding is not surprising however, since SAR131675 treatment inhibits the activity of VEGFR3 rather than its expression. We have included this explanation in section 3.4 of the revised manuscript.           

  1. Is there a change in PDL1 expression on tumor cells upon SAR131675 treatment.

In previous studies (Ardila et al 2020) we have shown that PDL1 was highly expressed on tumor cells and a different treatment did not alter this expression. This study did not investigate changes in tumor PDL1 expression following SAR131675 treatment.

  1. It may be too much for this study to investigate mechanism of VEGFR3 inhibition. However, some experiments like injecting tumor cells upon VEGFR3 deletion or VEGF-C deletion will help.

We acknowledge the reviewer’s comment, we agree with the reviewer that future experiments using VEGFR3 deleted tumor cells or VEGF-C inhibitors could delineate the mechanisms involved in tumor inhibition. We included this suggestion for future directions in the discussion of the revised manuscript. 

  1. To rule out contribution of angiogenesis, the authors should include angiogenesis inhibitors like bevacizumab (avastin) or VEGFR2 inhibitors as a control in their experiment.

We acknowledge the reviewer’s comment and respectfully explain that in previous studies (Alam et al 2012) demonstrated that SAR131675 is largely VEGFR3 specific. Additionally, in a previous study (Nguyen et al 2016), we used Sunitinib, a multi-kinase and VEGFR2 inhibitor, in this tumor model and demonstrated significant blood vessel reduction and tumor necrosis in a specific pattern. No such pattern was observed in the current study with SAR131675 treatment.  The text within the results section 3.2 and the discussion has been included to clarify this.

Reviewer 2 Report

Walsh and colleagues performed an in vivo study using an injection-based model of colorectal liver metastasis. They applied VEGFR3 targeting by a small molecule inhibitor SAR131675. This study does not cover a really novel topic, but they performed comprehensive immune compartment analysis using multicolor flow cytometry and likely identified an anti-tumor mechanism of VEGFR3 targeting in VEGFR3-positive tumors, which needs some further investigations. The following major points needs to be addressed:

  • The authors used CD103 as an activation marker of T cells. CD103 is not only expressed on T cells. Please state in the text, why this surface molecule was chosen. CD103 is known to directly bind to E-cadherin in tumors of epithelial origin. Since these tumors often undergo epithelial-to-mesenchymal transition (especially in metastases), it needs to be shown that tumors in the liver still express E-cadherin.
  • In all figures, findings that are not significant are not marked as “not significant”. Please include this information at least in the figure legends, that all significant data points are marked, and all others not. For example, in Figure 7 E, the reader might be not sure, if a * for significance is missing in the tumor group.
  • The Figures 6 and 7 need to show representative dot plots of the flow cytometry data. A general gating scheme in the supplementary figures is not enough.
  • The authors mentioned that no changes in lymph-/angiogenesis have been detected as a possible reason for the anti-tumor effect of SAR131675, although targeting of VGFR3 had this effect in other cancer types. They explained this finding by the shown VEGFR3 expression in colorectal liver metastases. However, they did not investigate whether the VEGFR3 ligand VEGF-C was present, as well. This is particularly interesting to the readers. IHC of tumors and ELISA of injected cells may be suitable to show this.
  • 300 µl volume of oral gavage in mice with a weight of 20g is quite much. Better indicate the oral gavage volume in ml/kg body weight.

Author Response

Walsh and colleagues performed an in vivo study using an injection-based model of colorectal liver metastasis. They applied VEGFR3 targeting by a small molecule inhibitor SAR131675. This study does not cover a really novel topic, but they performed comprehensive immune compartment analysis using multicolor flow cytometry and likely identified an anti-tumor mechanism of VEGFR3 targeting in VEGFR3-positive tumors, which needs some further investigations. The following major points needs to be addressed:

We thank the reviewer for their assessment of our submission and their constructive comments and suggestions, addressing these will improve the quality of the paper.

  1. The authors used CD103 as an activation marker of T cells. CD103 is not only expressed on T cells. Please state in the text, why this surface molecule was chosen. CD103 is known to directly bind to E-cadherin in tumors of epithelial origin. Since these tumors often undergo epithelial-to-mesenchymal transition (especially in metastases), it needs to be shown that tumors in the liver still express E-cadherin.

We acknowledge the reviewer’s comment, and agree that CD103 is  also expressed on cells other than T cells, including dendritic cells. In this study we specifically investigated CD103 expression on CD3+ T cells in order to determine the proportion of T cells with tissue resident status and whether the levels of these cells change by the SAR131675 treatment. The colorectal liver metastases in this model are strongly E-cadherin positive, while a small percentage of tumor cells exhibit mesenchymal morphology (Fifis et al 2013). We further clarified the use of CD103 marker in the revised manuscript modifying the text in section 3.7.

  1. In all figures, findings that are not significant are not marked as “not significant”. Please include this information at least in the figure legends, that all significant data points are marked, and all others not. For example, in Figure 7 E, the reader might be not sure, if a * for significance is missing in the tumor group.

We thank the reviewer for pointing out the omission, we have now corrected it. Figures legends 1-7 now contain statement “Significance is indicated by * while non-significance is unmarked” and tables 1-3 notes contain statement “Significance p<0.05 is indicated by bold text”.

  1. The Figures 6 and 7 need to show representative dot plots of the flow cytometry data. A general gating scheme in the supplementary figures is not enough.

We acknowledge the reviewer’s comment and have addressed these comments by modifying both Figure 6 and Figure 7 to include representative dot plots and histograms. Text within the results section 3.6 and 3.7 have been updated to reflect the modification of the figures. 

  1. The authors mentioned that no changes in lymph-/angiogenesis have been detected as a possible reason for the anti-tumor effect of SAR131675, although targeting of VGFR3 had this effect in other cancer types. They explained this finding by the shown VEGFR3 expression in colorectal liver metastases. However, they did not investigate whether the VEGFR3 ligand VEGF-C was present, as well. This is particularly interesting to the readers. IHC of tumors and ELISA of injected cells may be suitable to show this.

We acknowledge the reviewer’s comment, and we agree with the reviewer that VEGF-C staining could further shed light on the likely mechanism. We have added this suggestion in future directions in the revised manuscript.

  1. 300 µl volume of oral gavage in mice with a weight of 20g is quite much. Better indicate the oral gavage volume in ml/kg body weight.

We thank the reviewer for pointing this out. Oral Gavage of mice was performed in accordance with the supporting institution Animal Ethics Committee guidelines, where 300ul/20g is within the recommended rage of oral gavage administration. The manuscript has been modified to state that the dosage is 15ml/kg.

Round 2

Reviewer 1 Report

The authors did not address the concern raised by this reviewer experimentally (3 and 5).

Reviewer 2 Report

Please include the citation mentioned in the response letter (Fifis et al., 2013) to indicate that tumors from this cancer cell line originally had E-cadherin. The authors have adequately addressed the reviewers' points. 

Round 3

Reviewer 1 Report

accepted